# Whole-Body MRI Surveillance—Baseline Findings in the Swedish Multicentre Hereditary *TP53*-Related Cancer Syndrome Study (SWEP53)

**DOI:** 10.3390/cancers14020380

**Published:** 2022-01-13

**Authors:** Meis Omran, Emma Tham, Yvonne Brandberg, Håkan Ahlström, Claudia Lundgren, Ylva Paulsson-Karlsson, Ekaterina Kuchinskaya, Gustav Silander, Anna Rosén, Fredrik Persson, Henrik Leonhardt, Marie Stenmark-Askmalm, Johanna Berg, Danielle van Westen, Svetlana Bajalica-Lagercrantz, Lennart Blomqvist

**Affiliations:** 1Department of Oncology-Pathology, Karolinska Institute, SE-171 77 Stockholm, Sweden; yvonne.brandberg@ki.se (Y.B.); svetlana.lagercrantz@ki.se (S.B.-L.); 2Cancer Theme, Karolinska University Hospital Solna, SE-171 76 Stockholm, Sweden; 3Department of Molecular Medicine and Surgery, Karolinska Institute, SE-171 77 Stockholm, Sweden; emma.tham@ki.se (E.T.); lennart.k.blomqvist@ki.se (L.B.); 4Department of Clinical Genetics, Cancer Genetic Unit, Karolinska University Hospital Solna, SE-171 76 Stockholm, Sweden; 5Department of Surgical Sciences, Section of Radiology, Uppsala University, SE-751 85 Uppsala, Sweden; hakan.ahlstrom@radiol.uu.se; 6Department of Immunology, Genetics and Pathology, Uppsala University Hospital, SE-751 85 Uppsala, Sweden; claudia.lundgren@igp.uu.se (C.L.); ylva.paulsson@igp.uu.se (Y.P.-K.); 7Department of Clinical Genetics, Linköping University Hospital, SE-581 91 Linköping, Sweden; ekaterina.kuchinskaya@regionstockholm.se; 8Department of Radiation Sciences, Umeå University, SE-901 87 Umeå, Sweden; gustav.silander@regionvasterbotten.se (G.S.); anna.rosen@umu.se (A.R.); 9Department of Clinical Genetics and Genomics, Sahlgrenska University Hospital, SE-413 45 Gothenburg, Sweden; fredrik.l.persson@vgregion.se; 10Department of Radiology, Sahlgrenska University Hospital, SE-413 45 Gothenburg, Sweden; henrik.leonhardt@vgregion.se; 11Division of Clinical Genetics, Department of Laboratory Medicine, Office for Medical Services, Skåne University Hospital, SE-228 85 Lund, Sweden; marie.stenmark_askmalm@med.lu.se; 12Department of Translational Medicine, Radiology Diagnostics, Lund University, SE-205 02 Malmö, Sweden; johanna.berg@med.lu.se; 13Centre for Medical and Imaging and Function, SE-221 85 Lund, Sweden; danielle.van_westen@med.lu.se; 14Department of Diagnostic Radiology, Institution of Clinical Sciences, Lund University, SE-221 85 Lund, Sweden; 15Department of Imaging and Physiology, Karolinska University Hospital Solna, SE-171 76 Stockholm, Sweden

**Keywords:** cancer, cancer prevention, clinically actionable *TP53* variant, germline *TP53*, h*TP53*rc syndrome, Li–Fraumeni, hereditary breast cancer, hereditary cancer syndrome, MRI screening, surveillance program, whole-body MRI

## Abstract

**Simple Summary:**

Individuals who are born with a disease-causing variant of the *TP53* gene (hereditary *TP53*-related cancer syndrome, h*TP53*rc), also known as the Li–Fraumeni syndrome, have a very high (70–100%) lifetime risk of developing cancer and at younger ages. Carriers are also prone to develop secondary tumours due to irradiation. Current guidelines recommend surveillance programmes within studies and the use of non-irradiation modalities such as whole-body MRI (WB-MRI). In 2016, the Swedish *TP53* study (SWEP53) started inclusion, offering a surveillance program including WB-MRI. With this study, we aimed to describe the rate, anatomical distribution of malignant, indeterminate, and benign imaging findings as well as the associated further workup generated by the baseline WB-MRI in adult study participants. Our study identified the need of further workup in 19/61 participants, of whom three patients had a new cancer. WB-MRI appears to be a valuable surveillance strategy in families with h*TP53*rc syndrome.

**Abstract:**

A surveillance strategy of the heritable *TP53*-related cancer syndrome (h*TP53*rc), commonly referred to as the Li–Fraumeni syndrome (LFS), is studied in a prospective observational nationwide multi-centre study in Sweden (SWEP53). The aim of this sub-study is to evaluate whole-body MRI (WB-MRI) regarding the rate of malignant, indeterminate, and benign imaging findings and the associated further workup generated by the baseline examination. Individuals with h*TP53*rc were enrolled in a surveillance program including annual whole-body MRI (WB-MRI), brain-MRI, and in female carriers, dedicated breast MRI. A total of 68 adults ≥18 years old have been enrolled to date. Of these, 61 fulfilled the inclusion criteria for the baseline MRI scan. In total, 42 showed a normal scan, while 19 (31%) needed further workup, of whom three individuals (3/19 = 16%) were diagnosed with asymptomatic malignant tumours (thyroid cancer, disseminated upper GI cancer, and liver metastasis from a previous breast cancer). Forty-three participants were women, of whom 21 had performed risk-reducing mastectomy prior to inclusion. The remaining were monitored with breast MRI, and no breast tumours were detected on baseline MRI. WB-MRI has the potential to identify asymptomatic tumours in individuals with h*TP53*rc syndrome. The challenge is to adequately and efficiently investigate all indeterminate findings. Thus, a multidisciplinary team should be considered in surveillance programs for individuals with h*TP53*rc syndrome.

## 1. Introduction 

Carriers of a disease-causing germline *TP53* variant have a lifetime risk of 70–100% [1] of developing cancer of various types. This condition has commonly been referred to as the Li–Fraumeni syndrome (LFS) [2]. During recent years, it has become evident that these families can have different phenotypes—from the full LFS spectrum with childhood tumours and multiple cancer types to predominantly breast cancer in adults. Therefore, the European Reference Network for rare Genetic Tumour Risk Syndromes (ERN GENTURIS) recommends using the name heritable *TP53*-related cancer syndrome (h*TP53*rc) rather than LFS [3]. They also recommend that adult carriers should be surveilled yearly with whole-body magnetic resonance imaging (WB-MRI), since, at present, it is not possible to predict the tumour risk spectra for different families. 

Individuals with h*TP53*rc are at risk of developing cancer at considerably younger ages (median age 25 years) than non-carriers, and up to 50% of all carriers develop a tumour before the age of 30 [4,5]. Breast cancer is by far the most common tumour type, occurring in 30% of all female carriers [6]. *TP53*-carriers are more prone to develop multiple primary cancers [7]. The potential mutagenic effects of ionizing radiation are also higher in this group [8,9,10] which impacts the choice of imaging modality in the surveillance situation for healthy carriers. 

Persons with h*TP53*rc would likely benefit from surveillance to facilitate early detection of tumours in a potentially curative stage. In 2016, Villani et al. [11] published a pivotal 11-year follow-up of persons undergoing annual WB-MRI, showing a survival benefit for individuals undergoing imaging surveillance compared to non-surveillance. Historical cases were, however, used as controls, and the participants were enrolled primarily from families with a more severe phenotype including childhood cancers.

A previous meta-analysis of baseline WB-MRI as a cancer screening tool, including 578 carriers of a pathogenic *TP53* variant, reported a 7% detection rate of new, localized primary cancers [12]. The reasonable total acquisition time (roughly 60–90 min, depending on if breast-MRI is performed or not) without need for ionizing irradiation [13] makes WB-MRI suitable as a screening modality for individuals with germline *TP53* variants. To our knowledge, there are as yet no other ongoing studies in the Scandinavian countries, but studies with WB-MRI are reported in Australia [14], Brazil, Canada [11], Great Britain [15], the Netherlands [16], France [17], and the USA [18,19]. So far, little has been described regarding the incidence, anatomical distribution, and workup of imaging findings in this group of carriers.

The reported participants have been enrolled for WB-MRI within the Swedish *TP53* study (SWEP53), which aims to improve the clinical care of families with h*TP53*rc syndrome. The details of the SWEP53 study protocol have previously been described [20].

The aim of this report is to evaluate baseline WB-MRI regarding the distribution and rate of radiological findings in terms of malignant, indeterminate, and benign lesions and the associated further workup generated by this surveillance strategy. 

## 2. Materials and Methods

### 2.1. Recruitment of Study Participants

Individuals with pathogenic or likely pathogenic germline *TP53* variants were invited to participate in SWEP53 from 1 April 2016. Those who had performed their baseline WB-MRI by 1 May 2021 were included in this study. Inclusion was performed at the six cancer genetic units in Sweden, i.e., at the University Hospital in Umeå, Uppsala University Hospital, Karolinska University Hospital (Stockholm), Linköping University Hospital, Sahlgrenska University Hospital (Gothenburg), and Skåne University Hospital (Lund). Individuals were identified as carriers of a disease-causing *TP53* variant by one of these four testing procedures: (1) through gene panel testing within the clinical workup of participants with a suspected hereditary breast cancer, (2) through gene panel testing within a national research study aiming at identifying novel high risk genes for breast cancer, (3) targeted testing of *TP53* due to multiple primary cancers or family history fulfilling the Chompret 2015 criteria [4], or (4) through carrier testing of a healthy individual for a pathogenic *TP53* variant previously identified in the family. Written informed consent was obtained prior to inclusion. Ethical permission was obtained by the regional ethical review board in Stockholm with approval number 2015/1600-3 with the amendments 2017/1527-32 and 2018/1690-32. Exclusion criteria were individuals with general contraindications to MRI, any co-morbidity that precludes treatment of a cancer found in the surveillance program, or an ongoing intensive cancer treatment, where priority was given to clinical examinations.

The WB-MRI was defined as the baseline if it was either the first scan performed after study inclusion or, in some cases, a clinically performed WB-MRI prior to the inclusion date. 

### 2.2. Surveillance Program

The surveillance program included WB-MRI, a separate brain MRI, and in women with no previous bilateral risk-reducing mastectomy (RRM), breast MRI and breast ultrasound. All study participants were informed of the risk of possible secondary workup due to findings detected by the radiological and/or clinical examinations. 

### 2.3. Imaging Protocols 

The yearly WB-MRI examinations were intended to cover the whole body from the skull base to the feet (Figure 1). The SWEP53 study is nationwide, involving different clinical radiological units. Due to variations in the local set up at the different hospitals, the examinations were sometimes only extended to just below the patella or with more or less incomplete coverage of the upper extremities, such as in obese individuals where the distal parts of the upper limbs cannot be visualised. For men, and women who had undergone RRM, the WB-MRI and brain MRI were performed without intravenous (iv) contrast medium administration. For women without RRM, an iv-enhanced protocol was used when the examination was synchronized with breast MRI requiring contrast enhancement (Appendix A for further details). 

### 2.4. MRI

The study sites were provided with a protocol (Appendix A), and we held introductory meetings at all sites. The start of inclusion at each site took place between April 2016 and March 2020, starting with the Stockholm site. 

### 2.5. Evaluation

The evaluation of findings on whole-body MRI were single- or double read by radiologists and depending on need, local resources, and routines also separately evaluated by a neuro- and breast radiologist. For the purpose of reporting the findings on baseline WB-MRI, benign lesions were defined as those that were judged to be obviously benign on the MRI (and/or on prior imaging) requiring no further additional work-up. The clearly benign findings were handled as they would have been treated in a clinical setting. Findings referred to as indeterminate were those that required a further investigation to state whether benign or malignant. Malignant findings were defined as a new primary tumour, previously undiagnosed recurrence, or metastatic disease. Multiple lesions within the same organ (such as multiple liver metastases) were regarded as one finding.

### 2.6. Data Collection

Clinical data were collected from the medical records, including information on family history of cancer, previous cancer diagnosis and age at onset, findings at WB-MRI, and any subsequent workup. A typical workflow was inclusion at the site by an oncologist or a clinical geneticist. Physical evaluation and referral for WB-MRI followed, and results of the imaging were managed by the referring physician in accordance with established clinical procedures. The radiological reports and results of the workups from each participating site were then reviewed by the central site at Karolinska University Hospital in Stockholm. 

### 2.7. Statistical Analyses

All data presented are observational. Data are presented as numbers and proportions. 

## 3. Results

### 3.1. Previous Tumour Spectrum Prior to Inclusion

The mean age for the whole study population was 39 years (range 18–74 years old). Out of the 61 individuals, 32 had a previous history of tumour diagnosis prior to study inclusion (excluding ductal carcinoma in situ found in prophylactic mastectomies, malignant melanoma in situ, basalioma, and cervical cancer in situ). Breast cancer was the most common (23 participants with 27 breast cancer diagnoses). A total of 18 individuals had had 21 other tumour types. Tumour onset ranged from 4 months to 71 years of age. Thirteen participants had previously had more than one different type of malignancy by the time of study inclusion (Table 1).

### 3.2. Baseline Findings

By 1 May 2021, a total of 68 adult participants had been included nationally in SWEP53, of whom 61 adults had performed their baseline scans (Figure 2). 

A total of 30 new lesions (Table 2, Figure 3) were identified by WB-MRI in 19 individuals (31%) requiring further workup. Some participants had several lesions (Table 2). One participant had an indeterminate lesion on breast MRI but a benign diagnosis after ultrasound and biopsy.

#### 3.2.1. Malignant Findings

Overall, 9 of the 30 lesions (30%) that required further workup were malignant. All nine were identified in three individuals (3/61 = 5%): one participant had a new primary cancer, another had disseminated disease, and the third had a recurrent cancer. All patients were asymptomatic at the time of inclusion. 

#### 3.2.2. Indeterminate Findings

Of the 21 indeterminate findings (Figure 2, Table 2) requiring further workup in 19 individuals (19/61 = 31%), all investigations led to benign diagnoses, except one, where the participant had a pleural effusion on MRI. On follow-up with a CT scan, the effusion was regarded as physiological, but a primary papillary thyroid cancer was found instead. Notably, retrospective radiological evaluation of the WB-MRI did not detect the thyroid cancer, possibly due to the relatively low resolution of the WB-MRI and the limited number of image contrasts. Of the following workup, eight were invasive procedures such as biopsies or surgery (Table 2). 

#### 3.2.3. Benign Findings

In all 61 individuals, a total of 58 benign imaging findings were found, defined as benign by the radiologist without any need for further workup. The most common benign findings after further work-up were in the brain (pituitary adenomas and white matter signal alterations) and in the liver (benign lesions such as haemangiomas and focal nodular hyperplasia). 

## 4. Discussion

The results from this cohort of individuals with germline disease causing *TP53* variants showed that one-third of carriers had indeterminate imaging findings on baseline WB-MRI requiring further workup. Of these, all except one resulted in benign results. Of note, the rates of both indeterminate and benign imaging findings were higher than malignant findings, which were, in total, 9/88 lesions (10%) identified in 3/61 (5%) of all participants. A meta-analysis by Ballinger et al. in 2017 [12], comprising 578 participants with pathogenic germline *TP53* variants from 13 different cohorts who underwent baseline WB-MRI, reported 7% new malignancies. In contrast with our study, Ballinger et al. only included new localised cancers as new malignancies, whereas we defined previously unknown metastatic cancers and recurrences as malignant findings. We found new, non-metastatic malignancies in 1/61 (1.6%), thus lower than Ballinger et al. This is likely not due to age differences, as the mean age in our study group was 39 years (20–74, women) and 41 years (18–55, men) compared to Ballinger et al., who reported a mean age of 33 years. However, our cohort was smaller (61 compared to 578), and thus, the estimate should be interpreted with caution. We also included both persons with a classical LFS tumour spectrum and individuals from families with predominantly hereditary breast cancer. This could possibly have led to a lower detection rate due to an overall lower tumour risk for cancers other than breast cancer. In addition, we did not detect any cases of breast cancer in the 22 women who had not undergone RRM. This is not surprising, as the women were previously offered annual breast surveillance with breast MRI and ultrasound according to national guidelines. The fact that one patient had disseminated disease but a normal clinical examination highlights the difficulties in detecting cancer solely through clinical check-ups. We had a detection rate of 31% of indeterminate imaging findings requiring further investigation. In the meta-analysis by Ballinger et al. [12], the “false positives” were 42.5%. However, these were defined not only as lesions that were later found to be benign, but also included recurrences of pre-existing cancers and newly diagnosed metastatic cancers which could explain our lower rate of indeterminate lesions. The reported benign imaging findings might be considered to be high (58 lesions in 61 participants). However, this corresponds to the anticipated number of findings in a general clinical setting, as radiological reports often describe non-actionable lesions [21]. 

Although we had a defined study protocol for WB-MRI, some local adaptations were made due to different technical platforms, which led to slightly different anatomical coverage and imaging protocols. Since both soft-tissue sarcomas and osteosarcomas often occur in the limbs [22,23], it is vital to include the head to the toes including the extremities in the WB-MRI. According to our experience, this has to be conveyed to the radiology staff, since it has implications for the setup of the procedure and time needed for the examination. In addition, in many of the participating hospitals, the general introduction of WB-MRI as an examination was a challenge. It poses a new imaging modality that is often not routinely used in clinical practice. This must be in combination with (1) the challenge to review the large number of sequences and anatomical sections in an adequate manner, (2) the aim of detecting small lesions anywhere in the body, “needle in a haystack”, and (3) the importance of correctly evaluating any lesion in the context of the high tumour risk that is associated with germline *TP53* alterations. Instead of viewing the study participants as healthy individuals, perhaps the radiologists should consider them “patients with an unknown cancer” to minimize the risk of disregarding or overlooking potential malignant findings. 

In the future, it may be reasonable to exclude h*TP53*rc individuals with, for example, predominantly hereditary breast cancer from WB-MRI surveillance, and thereby reduce the risk of unnecessary workup. However, the benefits of repeated WB-MRI and risk stratification, potentially based on genotype–phenotype characterisations, is still under evaluation. Therefore, we suggest that all germline *TP53* carriers should be surveilled with WB-MRI in accordance with the ERN GENTURIS guidelines independent of family history [3].

## 5. Conclusions

Whole-body MRI in individuals with heritable *TP53*-related cancer syndrome may detect asymptomatic cancers. Benign and indeterminate imaging findings that require further work-up are more common than cancer. A multidisciplinary approach and a clinical infrastructure to manage these findings is needed to ensure adequate management.

## Figures and Tables

**Figure 1 cancers-14-00380-f001:**
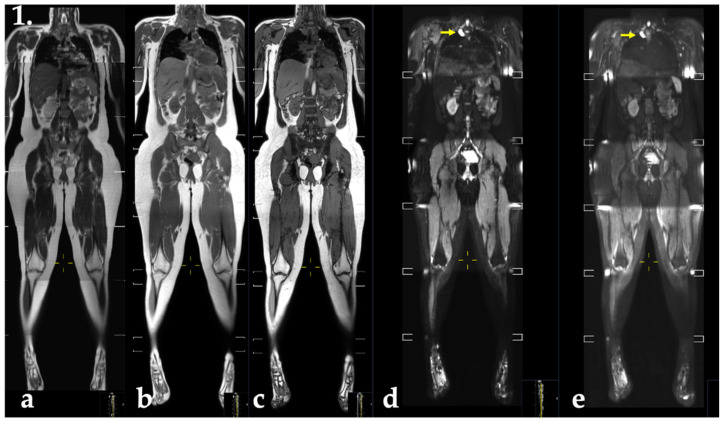
Display protocol of MR-imaging pulse sequences as specified in the SWEP53 study (female). Coronal stitched reformatted whole-body images with (**a**) T2-weighted (**b**) T1-weighted Dixon-based gradient-echo sequence with fat and water. (**c**) Opposed-phase (water and fat images no shown), echo-planar-based diffusion weighted images with b-values of (**d**) 50 s/mm^2^ and (**e**) 800 s/mm^2^. A 30 mm paravertebral thoracic lesion was found (**d**,**e**), yellow arrows, regarded as a schwannoma. Note the incomplete coverage of the distal part of the extremities on the whole-body images, in this case due to the individual’s size. The performed breast and brain MRI are not shown.

**Figure 2 cancers-14-00380-f002:**
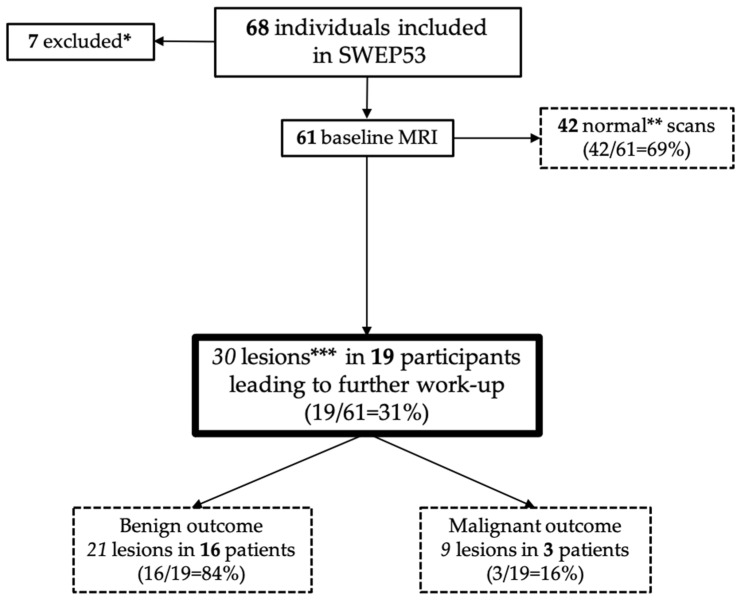
Flowchart of included individuals in SWEP53 and their imaging outcomes. * Two patients were not eligible for WB-MRI due to recurrent disease/new cancer diagnosis; two participants had not yet undergone WB-MRI. One person could not perform MRI due to pain; one withdrew consent for the study; one pending result. ** Normal scans = do not require any further workup. *** Lesions requiring further workup (imaging, fine needle aspiration cytology, biopsy, or referrals).

**Figure 3 cancers-14-00380-f003:**
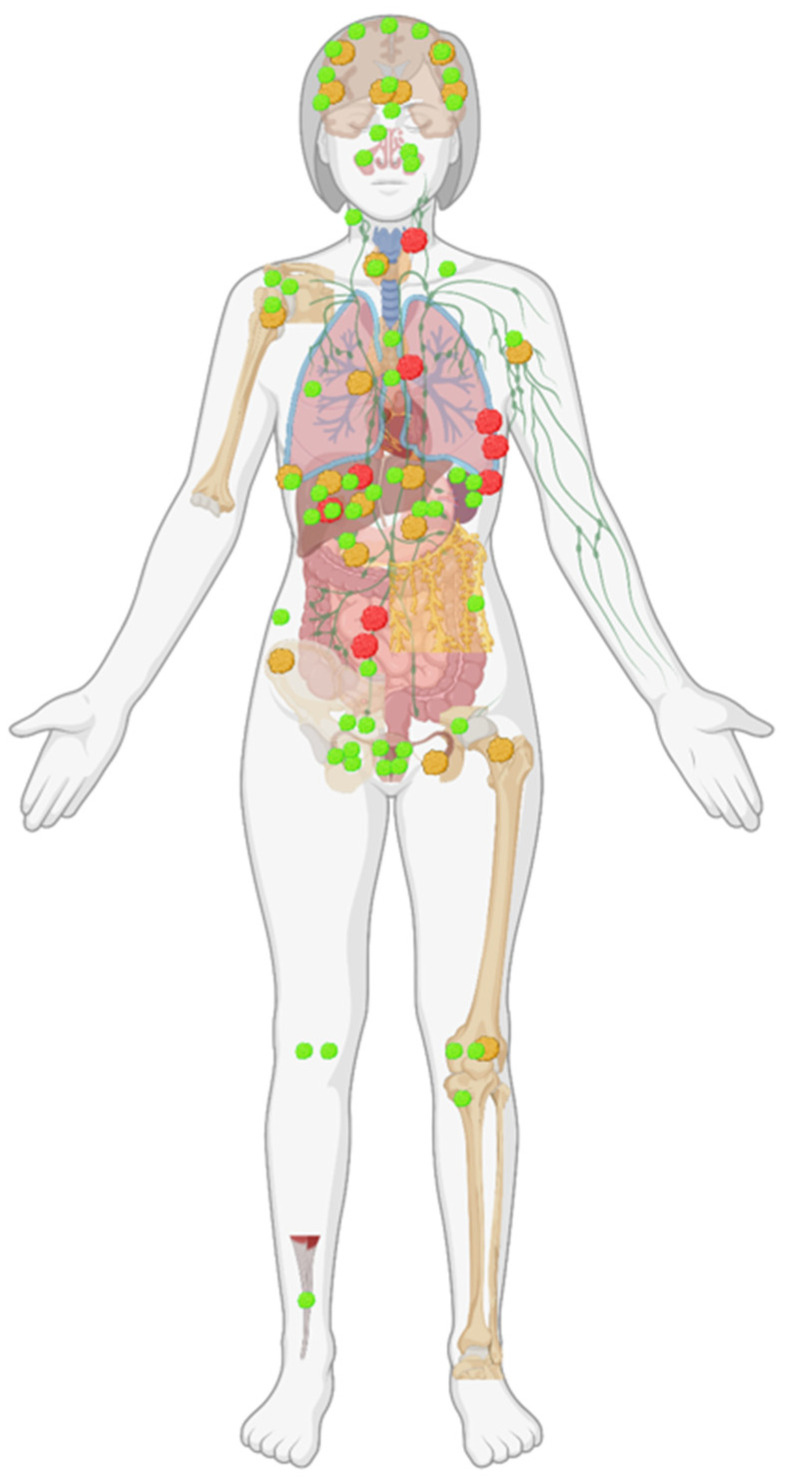
Anatomical distribution of baseline WB-MRI findings. Green indicating benign lesion, yellow indeterminate, and red indicating malignant findings. Image created with BioRender.com (accessed on 7 December 2021).

**Table 1 cancers-14-00380-t001:** Characteristics of participants within SWEP53 at inclusion.

Participant Characteristics at Inclusion	Women	Men
**No. of patients**	43	18
**Age** mean (range)	39 (20–74)	41 (18–55)
**Geographic region**		
Umeå	3	0
Uppsala	6	7
Stockholm	19	10
Linköping	3	0
Gothenburg	5	0
Lund	7	1
**Previous cancers**		
No tumours	15	14
Breast cancers	27 (23 patients)	0
Sarcomas	8 (6 patients)	1
Brain tumours	2 (2 patients)	1
Adrenocortical carcinomas	0	0
Other tumours *	7 (6 patients)	2 (2 patients)
**Multiple tumours ****		
1 tumour	16	3
2 different tumours	6	1
3 different tumours	6	0
4 different tumours	0	0
Risk-reducing mastectomy	21	0

* Other tumours: oesophageal carcinoma, lung adenocarcinoma, malignant melanoma (2), ovarian mucinous adenocarcinoma, Paget’s disease, prostate cancer, thyroid cancer, unclear mediastinal tumour. Overall, 29 individuals (15 women and 14 men) had no previous tumours prior to inclusion, while the others (32) had a history of one or several tumours. ** Multiple tumours indicating one individual having several different tumours, such as two different breast cancers.

**Table 2 cancers-14-00380-t002:** WB-MRI outcomes in the 61 participants.

WB-MRI Outcomes	Findings (*n* = Total Number of Findings)			Additional Workup (*n* = Individuals)		Result
Organ	Benign (Green)	Indeterminate (Yellow)	Malignant (Red)	Radiological	Other	
**Head, neck, arms**						
Brain	9	4	0	Brain-MRI * (4)	Referral to neurologist (1)	Benign
Pituitary	1	2	0	Brain-MRI * (2)	Referral to endocrinologist (2)	Benign
Face (subcutaneous)	1	0	0			Benign
Sinonasal cavity	3	0	0			Benign
Neck	0	0	0			
Lymph nodes, cervical	1	0	1		Part of workup for A	Benign **Malignant (A)**
Lymph nodes, axillary	1	1	0	Ultrasound	FNAC (1)	Benign
Arms	3	1	0	X-ray		Benign
Thyroid	1	1	0	Ultrasound (1)	FNAC (1)	Benign
Supraclavicular fossa	1	0	0			Benign
**Thorax**						
Lung	1	0	0			Benign
Lymph nodes, mediastinal	0	1	1	CT thorax-abdomen (1) CT thorax (1)	FNAC (2)	**Malignant (B)** Benign
Pleura	1	1	3	CT thorax (2) US thyroid (1)	FNAC thyroid (1)	**Malignant (B)** **Malignant (A)** Benign
Subcutaneous	0	0	0			
Vertebral column	2	0	0			Benign
**Abdomen**						
Peritoneum	1	0	0			Benign
Stomach	0	1	0		Gastroscopy (outside of study)	Benign
Lymph nodes, retroperitoneal	1	0	0			
Liver	7	3	2	Ultrasound (3) Liver-MRI * (2)	FNAC (2)	Benign (3) and **Malignant (B, C)**
Gall bladder	1	0	0			Benign
Pancreas	0	1	0	Pancreas-MRI*		Benign
Adrenal	0	0	0			
Kidneys	3	0	0			Benign
Spleen	3	0	0			Benign
Lymph nodes, intraabdominal	0	0	2	CT thorax-abdomen (2) Liver-MRI * (1)		**Malignant (B, C)**
Small bowel, colon, rectum and anus	0	0	0			
Subcutaneous	1	0	0			Benign
**Pelvis**						
Pelvic bone	0	1	0		FNAC	Benign
Lymph nodes, pelvic	0	0	0			
Uterus	4	1	0		Referral to gynaecologist (1)	Benign
Ovaries	3	1	0		Referral to gynaecologist (1)	Benign
Pelvic free fluid	2	0	0			
**Lower body**						
Legs	7	2	0	New imaging (1)	Operation and biopsy (1)	Benign Benign
**Total lesions: 88**	**58**	**21**	**9**	**24**	**15**	

All participants with indeterminate findings were women, except for one man with a thickening of the gastric wall. All three patients (A, B, C) with malignant (red) findings were women. A = papillary thyroid cancer. B = disseminated upper gastrointestinal cancer. C = recurrence of breast cancer. FNAC = fine needle aspiration cytology. Benign (green) and indeterminate (yellow). * With contrast enhancement.

## Data Availability

The data presented in this study are available upon request to the corresponding author. The data are not publicly available due to patient safety and confidentiality requirements.

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
