# Peer review of "Whole-Body MRI Surveillance—Baseline Findings in the Swedish Multicentre Hereditary TP53-Related Cancer Syndrome Study (SWEP53)"

_cancers, 2022, doi:10.3390/cancers14020380_

Round 1

Reviewer 1 Report

This manuscript does not add anything new to that previously published aside from a somewhat better annotation of MRI results requiring follow up.  That said, additional publications in this nascent field will add to the overall body of literature and this study does describe "real world" utilization of WB MRI including its limitations.   I would suggest the following.

  1. The authors imply that the term TP53-related cancer syndrome has essentially replaced the term Li-Fraumeni.   I would strongly disagree with this assertion.  
  2. There are small grammatical errors that should be fixed
  3. It is somewhat disconcerting that WB MRIs were performed as part of a study protocol, but despite this MRIs did not always include the entire extremeties despite the importance of this quality control aspect
  4. The outcome of case 3 with a malignancy is not clear to me.  It states disseminated breast cancer but discussed papillary thyroid cancer.  This requires clarification
  5. Detecting 1 localized thyroid cancer and 2 advanced metastatic cancers does illustrate to me the limitations of WB MRI as a screening test.   I think this should be emphasized in the discussion.  Many "false positives" that required investigation for very questionable survival benefit.  This could very well be related to the majority of cases identified through familial breast cancer cohorts.
  6.  The figures of brain MRI and breast MRI are not needed.   We have all seen many of these.  Okay to include WB MRI especially when it illustrates incomplete imaging of the extremities. 
  7. The "pivotal" Villani reference discussed is not referenced with a citation when first discussed. 

Author Response

Please se the attachment. 

Reviewer 2 Report

The author's present findings from a multi-institutional whole body MRI protocol in European patients.  This study is well written and highlights the findings from yet another cohort of patients to add to the growing body of literature.

Minor comments:

1. Perhaps the authors can also comment on how their findings relate to the recent controversial GENTURIS TP53 carrier mutation guidelines.

2. I would like to see the mutation profile of all the patients in the study.  Perhaps a supplemental table on the 75 patients could be added with genotype, personal history of cancer, family history of cancer and MRI findings.  Ideally perhaps some genotype-MRI finding guidance would be helpful, but likely not possible.
